# Passive Prey Discrimination in Surface Predatory Behaviour of Bait-Attracted White Sharks from Gansbaai, South Africa

**DOI:** 10.3390/ani11092583

**Published:** 2021-09-03

**Authors:** Primo Micarelli, Federico Chieppa, Antonio Pacifico, Enrico Rabboni, Francesca Romana Reinero

**Affiliations:** 1Sharks Studies Center—Scientific Institute, 58024 Massa Marittima, Italy; chieppafederico@gmail.com (F.C.); antonio.pacifico86@gmail.com (A.P.); ENRICO.RABBONI@gmail.com (E.R.); ricerca@centrostudisquali.org (F.R.R.); 2Department of Political Science and CEFOP-LUISS, LUISS Guido Carli University, 00197 Rome, Italy

**Keywords:** white shark, behaviour, Gansbaai, prey choice, shark vision, shark olfaction

## Abstract

**Simple Summary:**

White sharks, in surface passive prey predatory behaviour, are initially attracted by the olfactory trace determined by the bait and then implement their predatory choices to energetical richer prey, especially thanks to their visual ability, which plays an important role in adults and immatures with dietary shifts in their feeding patterns. Gansbaai represents a hunting training area for white sharks who are changing their diet.

**Abstract:**

Between the years 2008 and 2013, six annual research expeditions were carried out at Dyer Island (Gansbaai, South Africa) to study the surface behaviour of white sharks in the presence of two passive prey: tuna bait and a seal-shaped decoy. Sightings were performed from a commercial cage-diving boat over 247 h; 250 different white sharks, with a mean total length (TL) of 308 cm, were observed. Of these, 166 performed at least one or more interactions, for a total of 240 interactions with bait and the seal-shaped decoy. In Gansbaai, there is a population of transient white sharks consisting mainly of immature specimens throughout the year. Both mature and immature sharks preferred to prey on the seal-shaped decoy, probably due to the dietary shift that occurs in white sharks whose TL varies between 200 cm and 340 cm. As it is widely confirmed that white sharks change their diet from a predominantly piscivorous juvenile diet to a mature marine mammalian diet, it is possible that Gansbaai may be a hunting training area and that sharks show a discriminate food choice, a strategy that was adopted by the majority of specimens thanks to their ability to visualize energetically richer prey, after having been attracted by the odorous source represented by the tuna bait.

## 1. Introduction

The white shark (WS) *Carcharodon carcharias* [1] is an important top predator and the largest fish predator in existence, reaching about 6 m in length, combining many particular features including large size, regional endothermy (restricted to swimming muscles, viscera, and brain), and coarsely serrated dentition [2]. Predatory behaviour was recently described by Martin et al. [3], who provided ethograms with frequency and event sequence analyses of behavioural units for the Seal Island white shark population in South Africa, and by Hammerschlag et al. [4], who reported the effects of environmental factors on the frequency and success rate of predatory attacks. The white shark is an interesting species in the study of shark behaviour thanks to the relative ease with which it can be observed from the surface, especially near pinniped colonies on rocky islands where sharks congregate [5]. White sharks have the largest olfactory bulb among sharks, and thanks to their perceptions, they are able to trace wounded prey, whale carcasses, seal colonies or sea lions even at great distances [3]. During hunting, sharks refer mainly to odorous stimuli from prey [5], but many sharks are also thought to rely on their visual system for prey detection, predator avoidance, navigation, and communication [6]. For many sharks, vision plays a vital role in their ecology, particularly in detecting and identifying prey [7,8,9]. It is widely accepted that white sharks undergo an ontogenetic shift in prey preference [2,10,11,12,13]; both stomach-content and stable-isotope analyses indicate that an ontogenetic shift is expressed by a change in the trophic level, passing from a predominantly piscivorous diet when immature to a marine mammalian diet when adult [10,12,13,14,15]. The estimated total length (TL) at which they undergo this dietary shift varies between 200 cm and 340 cm [2,10,11,12,13,16,17]. White sharks’ teeth also reflect their ontogenetic diet change: as adults, the teeth of their upper jaw have a single large cusp, which is triangular and has serrated edges, while those of the lower jaw are tighter, smaller, narrower, and slightly sharper. The lower teeth penetrate and hold the prey, the upper ones cut the flesh: this enables both the predation of large prey such as pinnipeds, and the detachment of large pieces of meat from carcasses [18]. Martin et al. [3] observed that Sea Island white sharks appear to select the age class of Cape fur seal *Arctocephalus pusillus pusillus* [19], a group with a defined size and a clear direction of movement, as well as a choice of hunting during times and in locations that maximize their probability for predatory success. Regarding the ontogenetic shift, French et al. [20] suggested that gender and individual specialization are also key drivers in white sharks’ ecological variation, and that they remain important throughout ontogeny; in fact, individuals may learn a variety of different movement tactics for encountering and catching prey, and they may develop a preference for a particular tactic based on their experiences [21,22]. The individual surface behaviour of white sharks in the presence of a bait, in the same manner as their predatory [3,23,24] and social [25] behaviour, is not a simple stimulus response reflex, but rather a complex tactical situation with plastic responses [24]. Only a few studies have described the mechanisms that underlie patterns of prey selection in white sharks [9,26,27,28]. The present study carried out along the South African coasts in Gansbaai was aimed at providing a contribution to confirm whether, by simulating the natural condition of scavenging, white sharks, in the presence of artificial passive prey, such as a seal-shaped decoy and tuna bait, implemented a real food choice based on vision. In the attempt to gather new information on the topic, this study on the Gansbaai white shark transient population aimed to research the following: (i) the main surface predatory behaviours of white sharks when using both baits and seal-shaped decoys; (ii) the existence of possible interlinkages between predatory behaviour and other observed endogenous factors, such as maturity; and (iii) white sharks’ tendency to exhibit a real food choice, based on vision, rather than indiscriminate attacks on the two target passive preys, which are the bait and seal-shaped decoy.

## 2. Materials and Methods

Observations and data collection were recorded in the Dyer Island Natural Reserve, located 7.5 km south-east of Gansbaai, South Africa (34°41′ S; 19°24′ E). The reserve includes Dyer Island and Geyser Rock (Figure 1): the first is a low-profile island ca. 1.5 km long and 0.5 km wide, and it is characterised by the presence of different seabird colonies; the second is ca. 0.5 km long and 180 m wide, and it hosts a colony of Cape fur seals, *Arctocephalus pusillus pusillus* [19]. 

In Gansbaai (South Africa), at Dyer Island Nature Reserve, a large white shark population is present and can be observed thanks to the support of local ecotourism operators authorised to reach the field observation sites, with a prevalence of immature individuals [29]. The greatest proportion of sub-adults and potentially mature sharks also occurs in Seal Island, False Bay, South Africa [30,31]. The 2008–2013 study periods occurred in the autumn season between March and May and required a total of 247 h. Seal-shaped decoys and floating baits of tuna pieces, of similar size, were the tested passive target preys; in line with the research protocol adopted by Sperone et al. [24], the two baits were chosen first on the basis of the odourless seal shape, and second, the production of odour. The decoy’s size was as close as possible to that of a juvenile Cape fur seal (*A. pusillus pusillus*): 70 cm long and 32 cm wide, with a diameter of about 60 cm (Figure 2a,b).

The ethological observations on the sharks were made from the 13 m “Barracuda” boat owned by “Shark diving unlimited”, which was anchored at 100–150 m off Dyer Island. The boat was equipped with an upper deck from which it was possible to observe and photograph the sighted sharks. The observations were also carried out in an anti-shark cage, which was fixed, for the duration of the observations, to the side of the boat. Throughout this study, sharks were identified by the same research team (Sharks Studies Center—Centro Studi Squali—Istituto Scientifico, Massa Marittima, Italy), and the identification was based on the recognition of different anatomical features such as the dorsal fin, the caudal fin, and the presence of scars and ectoparasites and their arrangement [29,32,33]. It was also possible that one specimen was observed at various times throughout the day and that all exhibited behaviours were recorded. All sharks’ total lengths were estimated from the boat, referring to structures of a size known as the length of the cage. Inside the cage, operators were equipped with a mask, boots, a semi-dry suit and weights, remaining on the surface and maintaining a vertical position. When white sharks approached, the operators descended in a free dive to the bottom of the cage, in order not to disturb the sharks with air bubbles and to be able to observe, up close, the behaviour of the animals, the sex of the specimen and other body features. The cage was, therefore, an excellent and useful tool for identifying the sex of each individual with precision and for supporting the identification made on board. The sex of each shark was determined by the surface and by cage-diving observations, and also with underwater video recordings of the pelvic fin area: the males were recorded if claspers were seen and the females if the lack of claspers was verified and their pelvic fin area was filmed [29]. All other specimens were categorised as being of unknown sex. In our study, we estimated the white shark’s size at sexual maturity according to Hewitt et al. [31] and Micarelli et al. [29]: a mature male if the TL was ≥350 cm and a mature female if the TL was ≥450 cm. To attract the sharks, olfactory stimulants (chum) were used, following the methods described in Laroche et al. [34], Ferreira and Ferreira [35], and Sperone et al. [25]. The chumming was composed of sea water, cod liver oil (Gadus sp.), tuna blood and small pieces of fish [25,36]. Shark predatory behaviour was induced using two types of surface passive prey placed into the water: the bait, consisting of tuna pieces tied to a floating buoy positioned at the stern of the boat, and a seal-shaped decoy, positioned at the bow at a distance from the tuna bait of at least 10 m. A constant distance between the two-surface passive prey was maintained to isolate the seal-shaped decoy from the odorous and bloody trail coming from the tuna bait. In order to analyse the surface behaviour of white sharks in the presence of passive preys, tuna baits and an odourless seal-shaped decoy, we performed a Chi-squared independence test to check whether the types of prey (bait and seal-shaped decoy) were associated, and whether the choice of prey was not causal. To investigate the presence of independency between two causal variables, we used Pearson’s Chi-squared test, where H0 tests the null hypothesis of independency between the variables, and H1 the alternative hypothesis of dependency. To strengthen the results regarding the presence of causality between behaviour and prey, we also used Cochran’s Q test. This is a non-parametric statistical test used to verify whether k treatments (or number of studies) have identical effects. Generally, the test statistic refers to two-way randomized block designs, where the response variable takes only two possible outcomes coded as 0 or 1 denoting failure or success, respectively. It is often used to assess whether different observers of the same phenomenon have consistent results (interobserver variability). Cochran’s Q test is as follows: the null hypothesis (*H_0_*), where there is no difference in the effectiveness of treatments (the choice is causal) and the alternative hypothesis (*H*_1_), where there is a difference in the effectiveness of treatments (the choice is not causal). 

## 3. Results

### 3.1. Descriptive Analysis

Overall, 250 white sharks were sighted, with 166 having at least one or more interactions for a total of 240 interactions: 41 adults, 183 immatures and 16 unsexed. The variables denoting maturity and prey were transformed into dichotomous variables in order to be quantitatively evaluated (Table 1). In the statistical analysis, the re-interaction (more precisely, we observed 64 reinteractions) was counted as additional information investigating the frequency among white sharks’ individual characteristics. During the discriminant analysis, when studying the independency and heterogeneity among units, potential re-interactions were counted as unique observations. The same occurred in the empirical analysis accounting for non-linear methods, to investigate (potential) interactions among the observed variables. Multiple interactions were dealt with by shrinking the dataset with respect to every re-sighting. More precisely, the sample size ‘n’—used in the analysis—refers to the ‘individuals’ (counted only one time for any observation) and was weighted by the factors (e.g., behaviour and maturity). In this way, any (potential) bias—affecting the robustness of estimates—was sufficiently avoided in the statistical analyses. For the shrinking, a discriminant analysis involved in constructing hypotheses for a significance test on an individual was used before the estimation procedure.

### 3.2. Statistical Analysis

In the analysis of the white sharks (individual) and the two types of prey (bait and seal-shaped decoy) (Figure 3), the preference of white sharks for the seal-shaped decoy rather than the bait was proven through appropriate test statistics. More precisely, we ran a Chi-squared test (Table 2) and Cochran’s test (Table 3) rejecting any causality concerning the shark’s choice under the null hypothesis. In this way, the descriptive statistics were confirmed, and it was proven that the preference for the seal-shaped decoy was not causal. 

### 3.3. Frequency Distribution

In order to investigate potential relationships between white sharks and types of prey, the previous preliminary analysis was expanded to account for related frequency distributions. More precisely, the relationship was maturity–prey, highlighting possible interactions between each shark’s life stage and hunting patterns. The relationship maturity–prey highlighted that adult and even immature sharks preferred the seal-shaped decoy, confirming previous results. We performed a Chi-squared test with each frequency distribution accounted for. The main results, summarised in Table 2, highlight that the null hypothesis of independency can be rejected according to the whole sample. More precisely, the relationship between the types of prey (bait and seal-shaped decoy) is associated with a p-value close to zero (=0.00). This finding shows that their choice is not causal. In Cochran’s test, the treatments denote how the choice of different prey affects the sharks’ behaviour, as shown in Table 3. Let k −1 = 1 be the degrees of freedom, with k denoting the types of prey; as the computed p-value is lower than the significance level (α = 5%), one should reject the null and then accept the alternative hypothesis.

## 4. Discussion

*Carcharodon carcharias* is a “top predator” that has a wide distribution in temperate and tropical areas [5,18] with a higher concentration in eight spots around the world [18], including South Africa. Studies aimed at investigating the various methods of white sharks’ surface artificial prey approach or their food choices are scarce: between 1989 and 1992, Anderson et al. [32] examined the predatory behaviour of the Californian Farallon Islands sharks towards decoys with different shapes, with an approximate size of prey represented by pinnipeds and also including the reproduction of a sea lion shape, for a total of 159 h of observation. In that study, it was observed that vision played a major role in the approach and predation activities: the sharks were attracted, not by smell, electric fields or vibrations coming from the baits, but only by the presence of odourless prey on the surface. Strong [9] carried out an experiment comparing two floating shapes, one with the shape of a seal and the other with a square shape: during the first experiences with these preys, the sighted sharks showed a significant preference for the seal shape. He also stated that in his study there was little doubt that the sharks initially located the bait via olfaction, and vision was clearly used to orient their actual approaches, but its relative importance to most shark species remains, however, poorly understood. The present study carried out along the South African coasts in Gansbaai was aimed at providing a contribution to confirm whether, by simulating the natural condition of scavenging, white sharks, in the presence of artificial passive prey, such as a seal-shaped decoy and tuna bait, implemented a real food choice. Micarelli et al. [29] stated that, in Gansbaai, the white shark population was always made up of a prevalence of immature individuals, and that the mean TL recorded was 308 cm. It is important to remember that the estimated length at which white sharks undergo dietary shifts varies between 200 cm and 340 cm in TL [2,10,11,12,13,16,17]; between 2008 and 2013, only six specimens showed a total length <200 cm, of 250 specimens. Tricas and McCosker [14] showed a clear prevalence of an ichthyophagous diet in immature specimens, while adult sharks preferred marine mammals at Dangerous Reef, South Australia. Moreover, in our study, considering the tests performed to distinguish the adult from the immature individuals, it emerged that the adult sharks seemed to prefer the seal-shaped decoy, and the frequency of attacks on tuna bait by the adult sharks was not significantly greater than that of the immature ones. However, since the Gansbaai population of white sharks showed a mean TL of 308 cm, it is possible that the majority of Gansbaai white sharks’ transient population had already undergone the dietary shift, and this would explain why immature specimens also showed a similar interest towards the seal-shaped decoy (Figure 4). 

Regarding our question (which are the white sharks’ main surface predatory behaviours when using both baits and seal-shaped decoys?), it is possible to conclude from our studies that, at least as far as the behaviour of adult specimens is concerned, the same predatory trend observed near the Californian coasts emerged, but also included most immatures with dietary shift. White sharks must be selective when there is an abundance of food; therefore, according to the Optimal Foraging Theory [37], they prefer the most caloric sources over low-energy ones [38]. In response to the question “Do white sharks tend to exhibit a real food choice or indiscriminate attacks on the two passive prey (bait and seal shaped decoy)?”, it is possible to conclude that there was a real food choice rather than an indiscriminate attack, and that this strategy was adopted by the majority of specimens, helped by the ability to visualize the energetically richer preys, also with respect to the odorous source represented by the tuna bait. In this regard, it was possible to observe, in this study, that, in the contemporary presence of surface prey, characterized by different stimulating conditions, the white sharks preferred the odourless seal-shaped decoy rather than the tuna bait, which produces a strong odour stimulus. In only two of the six observation years, 2011 and 2008, we recorded the prevalence of the bait choice instead of the seal-shaped decoy. We suppose that this different behaviour, as proposed by Sperone et al. [39], could be linked to environmental factors, such as cloud cover and water visibility. This allows us to confirm that vision, in the Gansbaai transient population of white sharks, as suggested by Anderson [32] and Strong [9], plays an important role during the investigation and choice of prey, helping adults and immatures, which are in progress with dietary shift, to optimize the result of their predatory activity. 

## 5. Conclusions

It is possible to confirm, based on the data collected in the expeditions between 2008 and 2013, that the Gansbaai white shark transient population shows food preferences that are similar to those already observed in adult specimens in similar studies carried out along the Australian and Californian coasts [14,32]. The presence of a white shark population showing a mean TL of 308 cm, and only six specimens out of 250 showing a TL <200 cm, implies that for the majority of the Gansbaai white shark population, the change in diet has already occurred or is in progress and, therefore, the interest in energetically richer prey can be important for the overall population. Gansbaai, therefore, could represent a hunting training area for white sharks who are changing or have already changed their diet. It can be assumed that white sharks are initially attracted by the olfactory trace [5] determined by the bait and then implement their predatory choices to energetical richer prey, especially thanks to their visual ability, which plays an important role in adults and immatures with dietary shifts in their feeding patterns. The behaviour shown in the presence of the two preys examined in this work is, therefore, linked to the different dietary needs of white sharks in different stages of development. These needs are linked both to the modification of the dental system and to the different energy necessary in the immature and adult stages. As demonstrated by Laroche et al. [34] and Sperone et al. [24], the results presented in this article are based on the assumption that white sharks do not respond to the presence of the boat, but focus their attention mainly on the floating object, because the presence of a boat has very little influence on the white sharks’ behaviour. 

## Figures and Tables

**Figure 1 animals-11-02583-f001:**
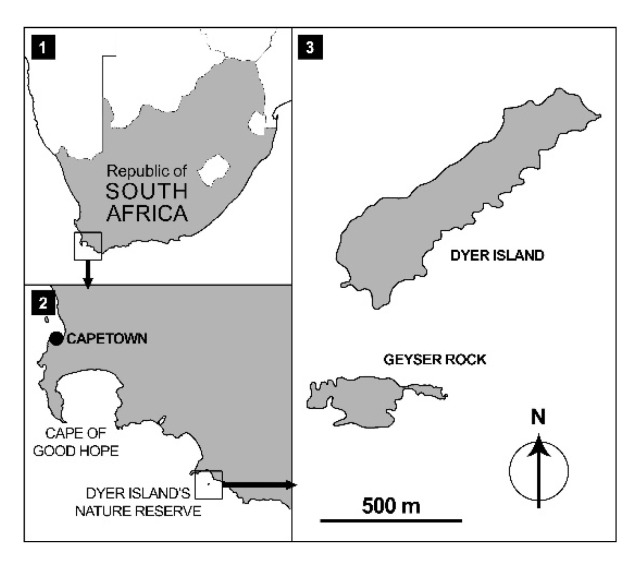
Dyer Island and Geyser Rock Nature Reserve.

**Figure 2 animals-11-02583-f002:**
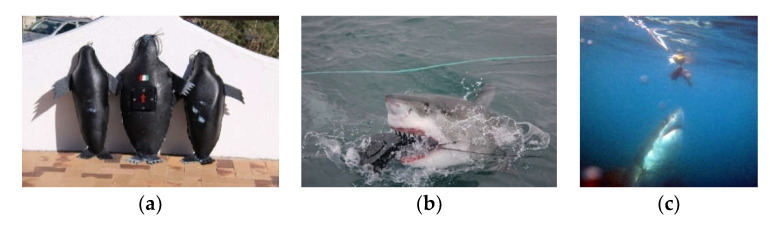
(**a**) Decoys, (**b**) interaction shark versus seal-shaped decoy, and (**c**) tuna bait.

**Figure 3 animals-11-02583-f003:**
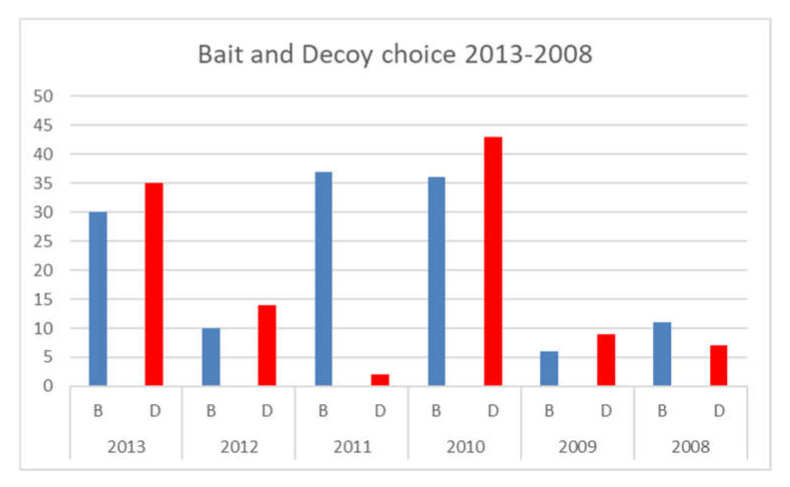
Bait and seal-shaped decoy shark choice, between 2013 and 2008.

**Figure 4 animals-11-02583-f004:**
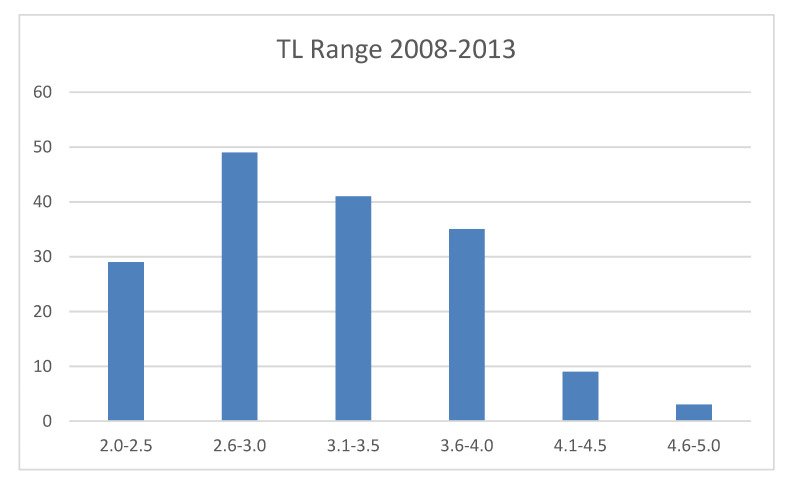
Total length range (meters) of white sharks, between 2008 and 2013.

**Table 1 animals-11-02583-t001:** Description of variables. Own computations.

Variable	Label
Maturity	(=1) adult, (=0) immature
Prey	(=1) bait, (=0) seal-shaped decoy

**Table 2 animals-11-02583-t002:** Pearson’s Chi-squared test of independence. The first column denotes the sub-categories, and the second column displays the corresponding statistical results in terms of *p*-values. The significant codes are as follows: * significance at 10%, ** significance at 5%, and *** significance at 1%.

Chi-Squared Test of Independence on Prey	
Full Sample	0.00 ***
Adult	0.00 ***
Immature	0.00 ***

**Table 3 animals-11-02583-t003:** Cochran’s Q test (asymptotic *p*-value).

Test Statistic	175.98
Chi-squared distribution (critical value)	3.840
Df	1
*p*-value (one-tailed)	0.027
Significance level	0.050

All the test statistics are significant (at 1%) and consistent (*p*-value < 0.001).

## Data Availability

https://www.researchgate.net/project/Great-White-Shark-Carcharodon-carcharias-Behaviour-Ecology-and-cotoxicology/update/61312b952897145fbd6df39c.

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
