# Peer review of "Passive Prey Discrimination in Surface Predatory Behaviour of Bait-Attracted White Sharks from Gansbaai, South Africa"

_animals, 2021, doi:10.3390/ani11092583_

Round 1
Reviewer 1 Report
The manuscript “Passive prey discrimination in surface predatory behavior of bait attracted white sharks from Gansbaai, South Africa” confirms some hypothesis on a poorly studied white shark’s behavior. However, the manuscript is often confused and needs to be revised. Furthermore, I have a major doubt regarding the analysis of multiple interactions.
Simple summary
Line 14
While the term “adults” can be linked to reproductive aspects, such as the sexual maturity, the term “young” is more generic. Please, consider replacing it with “immatures”, as it is in the ms
Abstract
Line 19
Are these 250 individuals different? Please specify it
Lines 21-27
Please, consider revising this part. The link between the dietary shift and the preferred prey needs to be clarified better. If a dietary shift occurs, then mature and immature individuals should have different preferences in feeding. Also, here it is not clear which is the link between the location (Gansbaai) and the hypothesis of this location as a training area. If sharks are attracted in this area by artificial baits, how can be hypothesized that they use it spontaneously for training? These issues are clearer after reading the work, but should be immediately understandable in the abstract.
Introduction
Lines 32-33
Please change “unusual” with “particular” or consider rephrasing. Large size and serrated dentition are features shared by many shark species
Lines 33-38
Please rephrase or split the sentence in two. The first part of the sentence “little is known about its surface predatory behavior” seems in contradiction with the second “it was described by…”
Lines 40-43
Please rephrase or split the sentence in two. It is not clear which is the link between the first part of the sentence “During hunting, sharks refer mainly to odorous stimuli from prey” and the second “in immature white sharks
Lines 43-46
I suggest moving this part to line 40, in order to introduce the argument
Lines 54-56
I suggest to link better this concept to those expressed from line 69
Materials and Methods
Lines 81-86
Please, add some information about the shark population observed in the area. Are there differences in the sharks observed around the two islands?
Lines 91-94
Please, clarify better why these two kind of baits were chosen (even if the protocol is well defined in Sperone et al., 2012, it is important to specify here this aspect).
Fig. 2
I suggest adding a tuna bait image and a picture showing how the bait and the decoy were positioned
Lines 134-149
Please, specify better here how multiple interactions were considered. If multiple interactions of the same individual were considered together in the same analysis, this compares them to single interactions carried out by different animals, maybe introducing some bias in the analysis. Moreover, it could be interesting to highlight if the same shark showed a similar behavioral pattern over time.
Lines 152-161
See previous comment
Lines 167 and 170
I think 3.2.1. should be deleted
Line 167
Please, rephrase: “larger” should refer to the number of sightings, but here it seems to refer to the size of the animals
Line 186
Please, consider including tab.2 values in the text, deleting the table
Discussion
Lines 215 – 216
This hypothesis should be reported also in the introduction and developed better.
Lines 200-233
This part is confused and needs to be clarified. Data from literature should be separated better from those belonging to the present work. Lines 226-229, I suggest to change the subject of the sentence from adult to immature individuals.
Conclusions
Lines 262-263
Please, develop better this statement.
Reviewer 2 Report
This is a worthwhile study representing a lot of field work but the manuscript need to be considerably reworked.
Some examples:
The Methods section needs to be revisited - There is confusion here about what the sharks were actually choosing between - Line 92 "floating baits of tuna pieces" or Line 129 - "2-3 Kg tuna". This is an important point because this whole experiment is about choosing between "natural" baits. It must be clearly defined as to what the choice really was - or was it sometimes a whole tuna and sometimes 'tuna pieces'
The section on statistical analysis has confusing language that makes it hard to decide what is actually going on. For example, Line 137 "...choice of prey was not casual" - Does casual here mean 'random' or is it a misspelling of 'causal"? And, in Line 138 "independancy between to causal variables". In this case is it really causal - or casual (random)? In short the paragraph of lines 134 - 149 needs to be reviewed and clarified.
For a fairly straightforward experiment, the Introduction is much too long and far ranging. It can be considerably shortened.
Reviewer 3 Report
The paper describes surface predatory behavior but does not indicate the depth range, particularly the maximum depth, of the species. This should be added, alongside other basic information such as the maximum size. Information and remarks on depth range and maximum size can be found in Weigmann (2016: Annotated checklist of the living sharks, batoids and chimaeras (Chondrichthyes) of the world, with a focus on biogeographical diversity. Journal of Fish Biology 88(3), 837–1037).
I appreciate the authors giving the species authorities for all taxa described. However, the new taxon descriptions (Linnaeus, 1758 etc.) should also be included in the list of references. Also, species authorities should be given when first mentioning of a taxon, i.e. species authority for Cape fur seal should be given in the Introduction instead of the Materials and Methods section.
The Materials and Methods indicates 2008–2013 (use n-dash instead of minus sign for all ranges) study periods but cites Martin et al. (2005) for the water temperature. This makes no sense.
It is stated that maturity stages were recorded following Hewitt et al. (2017). Although this is a reproducible method, it would be good to complement these findings by direct observations of male maturity based on the videos recorded and length of the claspers in relation to pelvic-fin length following e.g. Stehmann (2002: Proposal of a maturity stages scale for oviparous and viviparous cartilaginous fishes (Pisces, Chondrichthyes). Archive of Fishery and Marine Research 50(1), 23–48) and Tanaka et al. (2011: Age, growth and genetic status of the white shark (Carcharodon carcharias) from Kashima-nada, Japan. Marine and Freshwater Research 62, 548–556). In contrast to the 350 cm TL maturity threshold used for males in the present paper, Tanaka et al. (2011) indicated that males matured at 310 cm TL. These differences should also be discussed. See also discussion of individual variation in maturity size and comments on visual assessment of clasper length and rigidity from underwater video footage in Verkamp et al. (2021: Using reproductive hormone concentrations from the muscle of white sharks Carcharodon carcharias to evaluate reproductive status in the Northwest Atlantic Ocean. Endangered Species Research 44, 231–236).
The conclusion that the majority of specimens in the present study had already undergone the dietary shift based on total length is not sufficiently backed up by the data in my opinion. The authors state that the dietary shifts varies between 200 and 340 cm TL. As the majority of the sharks observed had total lengths within this range (according to Figure 3), it can only be speculated if the majority of sharks had already undergone the shift just based on total length. The preference of the decoy surely argues for a completed dietary shift in the majority of specimens, but this whole aspect should be discussed more concisely in the manuscript.
Author Response
REFEREE 3
ANSWER
Dear Referee,
thank you for your valuable and useful suggestions. We have tried to follow what you have proposed.
Inserted:
- Linnaeus, C. 1758. Systema naturae per regna tria naturae, secundum classes, ordines, genera, species, cum characteribus, differentiis, synonymis, locis. Ed. 10, Tomus 1. L. Salvii, Stockholm, Sweden, 823 pp.
- Schreber, J.C.D. Die Saugethiere in Abbildungen nach der Nature mit Beschreibungen. 1775. Wolfgang Walther, Erlangen, Vol. 2(13):223-230, pls 81-92.
We have removed the reference to Martin's temperature as required
The paper describes surface predatory behavior but does not indicate the depth range, particularly the maximum depth, of the species. This should be added, alongside other basic information such as the maximum size. Information and remarks on depth range and maximum size can be found in Weigmann (2016: Annotated checklist of the living sharks, batoids and chimaeras (Chondrichthyes) of the world, with a focus on biogeographical diversity. Journal of Fish Biology 88(3), 837–1037). ok
I appreciate the authors giving the species authorities for all taxa described. However, the new taxon descriptions (Linnaeus, 1758 etc.) should also be included in the list of references. Also, species authorities should be given when first mentioning of a taxon, i.e. species authority for Cape fur seal should be given in the Introduction instead of the Materials and Methods section. ok
The Materials and Methods indicates 2008–2013 (use n-dash instead of minus sign for all ranges) study periods but cites Martin et al. (2005) for the water temperature. This makes no sense. ok
It is stated that maturity stages were recorded following Hewitt et al. (2017). Although this is a reproducible method, it would be good to complement these findings by direct observations of male maturity based on the videos recorded and length of the claspers in relation to pelvic-fin length following e.g. Stehmann (2002: Proposal of a maturity stages scale for oviparous and viviparous cartilaginous fishes (Pisces, Chondrichthyes).
Relatively to: direct observations of male maturity based on the videos recorded and length of the claspers in relation to pelvic-fin length, we wrote; The cage was therefore an excellent and useful tool for identifying the sex of each individual with precision and for supporting the identification made on board. The sex of each shark was determined by the surface and by cage diving observations, and also with underwater video records of the pelvic fin area: the males were recorded if claspers were seen and the females if the lack of claspers was verified and their pelvic fin area was filmed (Micarelli et al. 2021). It’s impossible to meaure from the video single clasper length; sometimes it’s possible just in few specimens. Observe shark correctly placed in front of the cage happens rarely and, in this situation, it’s possbile to make a correct length evalutain of the claspers; it is easier to establish the maturity stage from the estimated total length, based on literature referencies.
Archive of Fishery and Marine Research 50(1), 23–48) and Tanaka et al. (2011: Age, growth and genetic status of the white shark (Carcharodon carcharias) from Kashima-nada, Japan. Marine and Freshwater Research 62, 548–556). In contrast to the 350 cm TL maturity threshold used for males in the present paper, Tanaka et al. (2011) indicated that males matured at 310 cm TL. These differences should also be discussed. See also discussion of individual variation in maturity size and comments on visual assessment of clasper length and rigidity from underwater video footage in Verkamp et al. (2021: Using reproductive hormone concentrations from the muscle of white sharks Carcharodon carcharias to evaluate reproductive status in the Northwest Atlantic Ocean. Endangered Species Research 44, 231–236).
Regarding the part on the maturity stage of white sharks, we know the article of Tanaka of 2011 and we did not refer specifically to this article as Tanaka itself says the following: Unique life-history traits, related to early growth rate and age at maturity, in Japanese white sharks may be caused by genetic differences from other populations. For this reason we felt that what was accepted and published both by us, Micarelli et al. 2021 and Hewitt et al. (2017), could be more appropriate for the South African population.
The conclusion that the majority of specimens in the present study had already undergone the dietary shift based on total length is not sufficiently backed up by the data in my opinion. The authors state that the dietary shifts varies between 200 and 340 cm TL. As the majority of the sharks observed had total lengths within this range (according to Figure 3), it can only be speculated if the majority of sharks had already undergone the shift just based on total length. The preference of the decoy surely argues for a completed dietary shift in the majority of specimens, but this whole aspect should be discussed more concisely in the manuscript.
At the end we did also correct the part relating to the conclusion.
Best regards

Round 2
Reviewer 1 Report
The authors improved the manuscript by following all suggestions and clarified my doubts about the way they handled the dataAuthor Response
Thank you.
Reviewer 2 Report
There are significant errors in the methodology and interpretation. For one thing, Yellowfin tuna do not inhabit waters less than 15 C and so they do not occur at the test location (Temperature 10 c) and would not have encountered the odor or site of a yellowfin for a long time - if ever. If the authors wish to demonstrate a preference for seals (and this is done on the basis of sight) the correct experiment would be to compare an artificial seal and and fish (of the appropriate i.e., similar sizes and locally available species). Sharks could be attracted to the boat using any smell stimulus but the choice must be between comparable targets. There is no description of the sequence of "investigation" versus "strike" by the animals - did the animals make investigations of both targets before choosing - or did they hit the first target they saw -or did they simply turn away? There is no documentation of the sequence of events or of the choice made by the sharks. T
The problems with the description of the analysis techniques have nit been addressed (e.g., "casual" versus "causal")
